# Heterobimetallic Ru(II)/M (M = Ag^+^, Cu^2+^, Pb^2+^) Complexes as Photosensitizers for Room-Temperature Gas Sensing

**DOI:** 10.3390/molecules27165058

**Published:** 2022-08-09

**Authors:** Abulkosim Nasriddinov, Sergey Tokarev, Vadim Platonov, Anatoly Botezzatu, Olga Fedorova, Marina Rumyantseva, Yuri Fedorov

**Affiliations:** 1Chemistry Department, Moscow State University, 119991 Moscow, Russia; 2Faculty of Materials Science, Moscow State University, 119991 Moscow, Russia; 3A.N. Nesmeyanov Institute of Organoelement Compounds RAS, 119991 Moscow, Russia

**Keywords:** heterobimetallic Ru(II) complexes, binding cations (Ag^+^, Pb^2+^ or Cu^2+^), nanocrystalline In_2_O_3_, hybrid materials, photoactivated gas sensing, selectivity

## Abstract

This work is devoted to the investigation of heterobimetallic Ru(II) complexes as photosensitizers for room-temperature photoactivated In_2_O_3_-based gas sensors. Nanocrystalline In_2_O_3_ was synthesized by the chemical precipitation method. The obtained In_2_O_3_ matrix has a single-phase bixbyite structure with an average grain size of 13–14 nm and a specific surface area of 72 ± 3 m^2^/g. The synthesis of new ditope ligands with different coordination centers, their ruthenium complexes, and the preparation of heterobimetallic complexes with various cations of heavy and transition metals (Ag^+^, Pb^2+^, or Cu^2+^) is reported. The heterobimetallic Ru(II) complexes were deposited onto the surface of the In_2_O_3_ matrix by impregnation. The obtained hybrid materials were characterized by X-ray fluorescent analysis, FTIR spectroscopy, and optical absorption spectroscopy. The elemental distribution on the hybrids was characterized by energy-dispersive X-ray spectroscopy (EDS) mapping. The gas sensor properties were investigated toward NO_2_, NO, and NH_3_ at room temperature under periodic blue LED irradiation. It was identified that the nature of the second binding cation in Ru(II) heterobimetallic complexes can influence the selectivity toward different gases. Thus, the maximum sensor signal for oxidizing gases (NO_2_, NO) was obtained for hybrids containing Ag^+^ or Pb^2+^ cations while the presence of Cu^2+^ cation results in the highest and reversible sensor response toward ammonia. This may be due to the specific adsorption of NH_3_ molecules on Cu^2+^ cations. On the other hand, Cu^2+^ ions are proposed to be active sites for the reduction of nitrogen oxides to N_2_. This fact leads to a significant decrease in the sensor response toward NO_2_ and NO gases.

## 1. Introduction

Nitrogen-containing gases such as NO_2_, NO, and NH_3_, are highly toxic to living organisms and can cause serious health damages if exposed for a long time [1,2,3,4,5]. The odor threshold value (OTV) for NH_3_ has a very wide range depending on the individual features of people and ranges from 0.04 to 57 ppm while its 8-h threshold limit value (TLV) is 25 ppm [6]. The OTV for NO_2_ ranges between 0.05 and 0.22 (0.4) ppm while the 8-h TLV is 0.5 ppm and the 15-min short-term exposure limit (STEL) is 1.0 ppm [7,8]. As can be seen, ammonia and nitrogen dioxide can be sensed by humans below the threshold; however, prolonged exposure to low concentrations can lead to adaptation and tolerance to the odor [9]. Moreover, the sad experience that covered the whole world in the face of the coronavirus (COVID-19) showed that such diseases can affect and seriously harm the human ability to sense smell and taste [10,11,12]. Olfactory neurons, in this case, may partially or completely lose their main function for an indefinite time. People who have been ill with coronavirus and work in industrial manufactures or laboratories and deal with toxic gases have a high risk of intoxication, due to the fact that their “natural sensor” is temporarily out of order. In addition, Exline et al., in their recent study, showed that the concentrations of biomarkers such as nitric oxide and ammonia are increased in patients with active COVID-19 pneumonia, which indicates the need for special non-invasive test systems for exhaled breathe analysis [13].

In such cases, metal oxide gas sensors, which are widely used for real-time monitoring of the composition of ambient air, can come to the rescue [14,15,16,17,18]. Through early detection of dangerous to human health trace concentrations of toxic gases in the case of leaks, they can notify users of the danger of being indoors in a timely manner. A promising approach for further development in this area and to obtain a more widespread distribution of sensing devices is to reduce the power consumption of the final product. The n-type metal oxide semiconductor indium oxide (In_2_O_3_), due to its relatively high electrical conductivity, has great potential for use as a gas sensor at low operating temperatures [19,20]. Earlier, it was shown that modification of the In_2_O_3_ surface with Ru(II) heteroleptic complexes as photosensitizers, on the one hand, allowed a reduction in the operating temperature down to room temperature using light illumination. On the other hand, it significantly increased the sensor response toward NO_2_ and NO [21,22,23]. It should be noted that the use of the separation of functions strategy, in which one part of the hybrid material is used for gas enrichment and the other part for direct amplification of the sensor signal, can have wide practical application [24], and the formation of a heterocontact can be effective for room-temperature NH_3_ and NO_2_ sensing [25,26,27].

On the basis of previous research [21,22,23], here, we synthesized new heterobimetallic Ru(II) complexes with various second binding cations (Ag^+^, Pb^2+^, or Cu^2+^) and used them as photosensitizers. The Ag^+^, Pb^2+^, and Cu^2+^ cations were chosen as promising for increasing the selectivity of gas analysis since they bind nitrogen-containing or sulfur-containing gases [28,29,30,31,32,33]. Tokarev et al. already showed that in such bimetallic systems, there is no charge transfer between cations, thus each of them has its own role: Ru(II)-containing fragment is a photosensitizer, and the second cation is a catalytic center or additional adsorption site in the interaction with the gas phase.

Thus, in the presented work, we studied the influence of the second binding cation on the light-activated gas sensor properties of the hybrids based on nanocrystalline In_2_O_3_ and Ru(II) heterobimetallic complexes. It was concluded that the nature of the second cation in Ru(II) heterobimetallic complexes can influence the selectivity toward different gases.

## 2. Results and Discussion

Analysis of the structural properties of the obtained nanocrystalline In_2_O_3_ showed that it has a single-phase bixbyite structure with an estimated average grain size of 13–14 nm (Appendix A). The specific surface area was 72 ± 3 m^2^/g.

Figure 1 represents the SEM image of the In_2_O_3_ sample (a) and EDS mapping of hybrid materials (b, c, d). It can be observed that In_2_O_3_ has a porous microstructure with uniformly aggregated nanoparticles. Moving from the scale of 200 nm to 10 µm, one can see that the porous morphology is well preserved. The results of the EDS mapping clearly indicate the homogeneous distribution of the Ru and second cation (Pb, Ag, or Cu) on the surface of the In_2_O_3_ matrix. The samples have similar particle sizes and porous structures, providing a large surface area.

The data on the elemental composition of the hybrids are given in Table 1. It can be seen that the content of the Ru in hybrids is close to the specified value of 0.7 mol%. These values were averaged from four different areas of the sample. The samples in this case had the optimum resistance to stay within the measuring range of the equipment while maintaining a high sensor signal.

Figure 2a shows the FTIR spectra of the heterobimetallic Ru(II) complexes in the frequency range of 4000–400 cm^−1^. The characteristic IR spectra of the complexes are similar. The broad peak in the range of 3200–3700 cm^−1^ is attributed to the stretching vibration of the N–H and O–H groups. The bands located between 2800 and 3100 cm^−1^ are associated with stretching C–H vibrations. The sharp and intense peaks in the range of 1000–1700 cm^−1^ are attributed to bpy ring breath, C=O, C=C, C–H, and C=N vibrations. The low-frequency region is assigned to the ρ(CH_2_), γ(CH), and ν(C–C) modes. However, in some studies in which both experimental and theoretical calculations were carried out [34,35,36], these sharp peaks, the values of which are close to those in this work, were attributed to the stretching vibration of the Ru–N bond while in other works, these frequencies were located lower. The corresponding vibration modes of the functional groups are shown in Table 2.

Figure 2b represents the FTIR spectra of the pure indium oxide and hybrid materials. All spectra contain intense broad signals corresponding to the stretching vibrations of the In–O bonds: the bands at 420 and 565 cm^–1^ correspond to symmetric and the band at 600 cm^–1^ to asymmetric stretching of the In–O valence bond. The spectra show absorption in the regions of 1625 cm^−1^ and a wide band with a maximum at 3432 cm^−1^, which refer to the bending vibrations of adsorbed water and stretching vibrations of surface OH groups, respectively. Weak signals from Ru(II) heterocyclic complexes are also observed in the range of 900–1690 cm^−1^.

The normalized optical absorption spectra of the heterobimetallic Ru(II) complexes, unmodified In_2_O_3_, and hybrids are shown in Figure 3.

There are several absorption bands in the spectra of the organic complexes (Figure 3a): the bands located in the ultraviolet region correspond to ligand-centered π–π* transitions in 2,2′-bipyridine ligand (*λ*_max_ = 288 nm) and imidazophenanthroline ligand (*λ*_max_ = 333 nm) [33,38]. The last one is intense only for the RA complex, which may be associated with a lower silver charge (+1 compared to +2 for copper and lead), while for the other two complexes (RC and RP), it appears as a shoulder. A broad band (*λ*_max_ = 460 nm) in the visible region is associated with the metal to ligand charge transfer (MLCT) transition, which plays a key role in photosensitization. This is why a blue LED was used as the illumination source.

The absorption spectrum of the pure In_2_O_3_ matrix does not have any bands in the visible region. The observed absorption band in the UV region corresponds to a direct transition from the valence band to the conduction band of the semiconductor. The wide absorption band in the visible region of the spectrum of hybrid materials corresponds to the MLCT transition in the organic part. Moreover, the maximum of the absorption edge is shifted to the longer wavelength by about 10 nm, thus it completely coincides with the emission spectrum of the blue LED.

Since sensor measurements were performed under periodic (blinking) light irradiation [20,21], there are two resistance values corresponding to dark and light conditions, which were used to characterize the gas response and photoresponse of the materials. The sensor signal toward oxidizing gases (NO and NO_2_) was calculated as a ratio of the “dark” resistance in gas atmosphere to the “dark” resistance in pure air atmosphere (Equation (1)); for the reducing gas (NH_3_), it has the reversed form (Equation (2)). Photoresponse was calculated as a ratio of the “dark” resistance to the “light” resistance consequently in air and testing gas atmosphere (3):(1)$$S=\frac{R_{dark}(gas)}{R_{dark}(air)}$$
(2)$$S=\frac{R_{dark}(air)}{R_{dark}(gas)}$$
(3)$$S_{ph}=\frac{R_{dark}}{R_{light}}$$

Figure 4a shows the dependence of the sensor’s resistance on the NO_2_ concentration. It can be seen that the sensitization of nanocrystalline indium oxide by organic complexes leads to an increase in the baseline resistance of the sensors, with the greatest effect being observed for the RC complex. Additionally, sensitization leads to an increase in the photoresponse, but the sensor signal increases only for the In_2_O_3_ + RA and In_2_O_3_ + RP hybrids while sensitization by the complex RC leads to a decrease in the sensor signal (Figure 4b,c).

As mentioned above, the illumination of the hybrids with blue LED leads to the MLCT transition in the organic part (Equation (4)). Then, the photoexcited electron can be transferred into the indium oxide’s conduction band. In pure air atmosphere when the LED is turned off, the resistance begins to increase due to oxygen adsorption accompanied with electron capture from the conduction band (Equation (5)). Illumination reduces the resistance of materials due to oxygen photodesorption during its interaction with photogenerated holes (Equation (6)). In NO_2_ atmosphere, the resistance begins to additionally increase due to the Reactions (7) and (8) in dark conditions. When the LED is turned on, desorption of NO_2_ (Reaction (9)) will also proceed along with Reaction (6):(4)$$hv\overset{\mathrm{MLCT}}{\to }h^{+}+e^{-}$$
(5)$$O_{2\left(\mathrm{gas}\right)}+e^{-}\to O_{2\left(\mathrm{ads}\right)}^{-}$$
(6)$$O_{2\left(\mathrm{ads}\right)}^{-}+h^{+}\left(hv\right)\to O_{2\left(\mathrm{gas}\right)}$$
(7)$${\mathrm{NO}}_{2\left(\mathrm{gas}\right)}+e^{-}↔{\mathrm{NO}}_{2\left(\mathrm{ads}\right)}^{-}$$
(8)$${\mathrm{NO}}_{2\left(\mathrm{gas}\right)}+O_{2\left(\mathrm{ads}\right)}^{-}↔{\mathrm{NO}}_{2\left(\mathrm{ads}\right)}^{-}+O_{2\left(\mathrm{gas}\right)}$$
(9)$${\mathrm{NO}}_{2\left(\mathrm{ads}\right)}^{-}+h^{+}\left(hv\right)↔{\mathrm{NO}}_{2\left(\mathrm{gas}\right)}$$

The gas sensor properties were also investigated towards NO in the concentration range from 0.5 to 4 ppm. The nature of the change in resistance was the same as in the interaction of sensors with nitrogen dioxide: resistance increases with increasing NO concentration. In an earlier work [22], it was shown by the DRIFTS method that at room temperature NO is oxidized by chemisorbed oxygen to NO_2_ and further reacts as an oxidizing gas according to the reaction (10). A schematic illustration is shown in Figure 5.
(10)$${\mathrm{NO}}_{\left(\mathrm{gas}\right)}+1/2O_{2\left(\mathrm{gas}\right)}+e^{-}↔{\mathrm{NO}}_{2\left(\mathrm{ads}\right)}^{-}$$

The study of the sensor properties toward ammonia was carried out in the concentration range of 5.0–20.0 ppm NH_3_ in air. The change in the resistance of the samples with a cyclic change in the composition of the gas phase is shown in Figure 4a. In the presence of NH_3_, the resistance of the samples decreases, which is consistent with the fact that the interaction of ammonia (reducing gas) with the surface of an n-type semiconductor leads to a decrease in resistance and the sensor signal increases with the increasing NH_3_ concentration (Figure 6b).

The value of the photoresponse characterizes the change in resistance during periodic illumination due to photodesorption and photoadsorption primarily of oxygen. In the case of NO_2_ (Figure 4c), an increase in the photoresponse was observed with the increasing concentration, which is associated with the additional contribution of the adsorption and desorption of nitrogen dioxide molecules, which has a high electron affinity. On the contrary, with an increase in the ammonia concentration, a decrease in the photoresponse was observed for all materials (Figure 6c). When the surface of the sensors interacts with ammonia, it is oxidized by oxygen chemisorbed on the surface of nanocrystalline material according to the following Reaction (11). Thus, due to the photodesorption of oxygen and its consumption (Reaction 6), at high ammonia concentrations, the amount of oxygen on the surface is very small.
(11)$$2\beta \cdot {\mathrm{NH}}_{3\left(\mathrm{gas}\right)}+3O_{\beta \left(\mathrm{ads}\right)}^{-\alpha }\;\to \;\beta \cdot N_{2\left(\mathrm{gas}\right)}+3\beta \cdot H_{2}O_{\left(\mathrm{gas}\right)}+3\alpha \cdot e^{-}$$

Moreover, as it turned out, the time in pure air flow (60 min) was not enough to completely restore oxygen on the surface. This is why for the In_2_O_3_, In_2_O_3_ + RA, and In_2_O_3_ + RP samples, there is a strong baseline drift and a slight change in the sensor signal. However, the In_2_O_3_ + RC sample behaves differently: its photoresponse has a maximum value in the case of detection of both oxidizing and reducing gases (Figure 7b). This is likely related to the structure of the RC complex. The Cu^2+^ cation is coordinated in the heterobimetallic Ru(II) complex via the azadithia-15-crown-5 fragment, which has a donor nitrogen atom. The interaction of the Cu^2+^ cation with the heteroatoms of the crown ether fragment can lead to the neutralization of the electron-donor function of nitrogen directly bound to the chromophore parts of the molecules [33]. It is likely that periodic photoactivation can make this process reversible.

Figure 7 represents the comparison of the sensor signal and photoresponse of the samples towards NO_2_, NO, and NH_3_ at room temperature.

As it can be observed, the In_2_O_3_+RC sample has the highest sensor signal toward NH_3_ (Figure 7a), and its resistance reversibly changes in the gas atmosphere (Figure 6a). These results can be explained in several ways. In terms of acid-base properties, copper is the stronger Lewis acid compared to silver and lead. This may lead to an increase in the sensitivity of the copper-containing hybrid material to ammonia. A number of papers showed the specific adsorption of NH_3_ molecules on copper cations [39,40,41,42,43,44,45]. The formation of the $[\mathrm{Cu}{\left({\mathrm{NH}}_{3}\right)}_{2}^{+}]\;$complex was shown by the DRIFTS method for a CuBr sample [39]. The formation of $[\mathrm{Cu}{\left({\mathrm{NH}}_{3}\right)}_{4}^{2+}]\;$complex is proposed to form at room temperature, and it was proved by several analysis methods [40]. Liang X. et al. calculated the binding energies of the M(II)–N bond in different M(II) phthalocyanine/MWCN hybrids with NH_3_ molecules at room temperature and concluded that the formation energy of the stable structure decreases in the following order: CoPc-NH_3_ > ZnPc-NH_3_ > CuPc-NH_3_ > PbPc-NH_3_ > PdPc-NH_3_ > NiPc-NH_3_, while Pb, Pd and Ni can facilitate the response of oxidants, such as NO and NO_2_ [28].

Moreover, Cu^2+^ ions are proposed to be an active site for NO decomposition or reduction of nitrogen oxides to N_2_ (Reaction (12)) [40]:(12)$$2{\mathrm{NO}}_{\left(\mathrm{ads}\right)}↔N_{2\left(\mathrm{gas}\right)}+{\;O}_{2\left(\mathrm{gas}\right)}$$

It can be supposed that the mechanism of the NO decomposition is more complicated and can precede the appearance of the NO_2_^δ−^ intermediate [40]. After NO dissociative adsorption, the Cu^2+^ sites can actively transform NO into 1/2N_2_; however, the released 1/2O_2_ will more probably react with another NO molecule to form [Cu^2+^-NO_2_^δ−^] rather than combine with another 1/2O_2_ and desorb as O_2_. Then, NO_2_^δ−^ can rearrange with NO^δ+^ to form [Cu^2+^-N_2_O_3_], which immediately decomposes into N_2_O + O_2_ or N_2_ + O_2_.

Since reaction (12) proceeds without electron exchange, it, therefore, does not contribute to the change in conductivity. However, a slight increase in the sensor signal is still observed and it is most likely associated with the adsorption of a part of the NO molecules on the active sites of the In_2_O_3_ matrix and further oxidation by chemisorbed oxygen according to Reaction (10). This can be evidenced by the fact that the sensor signal to NO for the In_2_O_3_ + RC hybrid is even lower than for pure In_2_O_3_. The same situation can be observed for NO_2_ detection (Figure 7a). However, as can be observed, for samples In_2_O_3_ + RA and In_2_O_3_ + RP, the sensor signal toward NO and NO_2_ is practically similar. In this case, the main role in detection is played by the ruthenium part of the photosensitizer. As a result of photoinduced charge transfer, the surface of the semiconductor is enriched with electrons, which enhances the interaction with oxidizing gases—electron acceptors.

Since the humidity of the surrounding atmosphere significantly affects the sensor properties of the gas detectors at room temperature, the effect of the relative humidity (RH) on the response to investigated gases was studied. Thus, the sensor properties were also tested when RH was 30%. Figure 8 shows that the presence of water vapor differently affected the sensor response compared to the tests in dry air.

Since the kinetic diameter of the water molecules (2.65 Å) is less than that of NH_3_ (2.9 Å), NO_2_ (3.4 Å), NO (3.17 Å), and O_2_ (3.46 Å), they can diffuse more easily through the porous structure of In_2_O_3_ and occupy active sites [46,47,48]. This condition adversely affects the interaction with ammonia, since the amount of chemisorbed oxygen capable of oxidizing ammonia molecules is significantly reduced, which resulted in a decrease in the sensor signal. For all samples, a decrease in the base resistance was found due to an increase in the number of charge carriers in the conduction band, which proceeds according to the proposed Reactions (13)–(15) [49,50]:H_2_O_(gas)_ + In_(lat)_ + O_(lat)_ = [In_(lat)_-OH] + [O_(lat)_H] + e^−^(13)
H_2_O_(gas)_ + 2In_(lat)_ + O_(lat)_ = 2[In_(lat)_-OH] + V_O_∙+ 2e^−^(14)
H_2_O_(gas)_ + 2In_(lat)_ + O_2_^−^_(ads)_ = 2[In_(lat)_-OH] + e^−^(15)

In the case of detecting NO_2_, an increase in the sensor response was observed due to the capturing of released electrons by nitrogen dioxide molecules in Reactions (13)–(15). The obtained results are in good agreement with the previously obtained data in the case of the use of a heteroleptic ruthenium(II) complex without a second cation as a photosensitizer [51]. However, in this work, the sensor signal of the hybrid materials toward NO also increased in a humid atmosphere. Most likely, the cations of the second metals contributed to the oxidation of NO molecules by OH groups.

The long-term stability of the sensors was investigated for 30 days toward 1 ppm NO_2_, 1 ppm NO, and 20 ppm NH_3_ at room temperature under periodic light illumination (Figure 9). The sensitive layer of the sensors was cleaned before each measurement for 2 h by blowing purified air under periodic illumination. The sensor signals remained fairly stable when detecting nitrogen oxides, demonstrating reproducibility and reversibility. When detecting ammonia, only the In_2_O_3_ + RC sample exhibited a stable signal with smaller error bars. It should be noted that the In_2_O_3_ + RA and In_2_O_3_ + RP samples showed the same signal path in all cases. However, it should be noted that during the sensor studies, over time, the reaction products accumulated on the surface of the materials and they were not completely desorbed at room temperature, which can significantly affect the durability of sensors.

After reviewing the literature, a comparative analysis of the obtained data of gas sensor measurements with similar works was carried out; the results are represented in Table 3. It should be noted that there are few works related to photoactivated sensors for the detection of ammonia and nitric oxide while there are a sufficient number of those for the detection of nitrogen dioxide at room temperature. The obtained samples in this work showed significant sensing characteristics at room temperature under photoactivation. Among the considered materials from the literature, the samples in this work did not show the outstanding values of the gas sensor characteristics. Nevertheless, it is worth noting that the response to ammonia is acceptable because the In_2_O_3_ + RC sample was capable of detection below the 8-h TLV. The In_2_O_3_ + RA and In_2_O_3_ + RP samples are inferior in terms of the response to NO and NO_2_ only to hybrid materials that were previously studied in our scientific group. However, a significant advance was observed in enhancing the sensor signal toward NO detection in humid air, which is expected to be promising in the analysis of biomarkers of exhaled air.
(16)$$S=\frac{R\left(\mathrm{air}\right)-R\left(\mathrm{gas}\right)}{R\left(\mathrm{gas}\right)}\;\times \;100\%,\;\mathrm{for}\;{\mathrm{NH}}_{3}$$
(17)$$S=\frac{R\left(\mathrm{gas}\right)-R\left(\mathrm{air}\right)}{R\left(\mathrm{air}\right)}\;\times \;100\%,\;\mathrm{for}\;\mathrm{NO}\;\mathrm{and}\;{\mathrm{NO}}_{2}$$

## 3. Materials and Methods

### 3.1. Materials Synthesis

#### 3.1.1. Synthesis of Nanocrystalline In_2_O_3_

Nanocrystalline In_2_O_3_ was synthesized by the chemical precipitation method and was used as an inorganic matrix in the hybrid materials. The synthesis method and results of the structural characteristics of In_2_O_3_ were discussed in detail in our earlier work [22]. Briefly, the In(OH)_3_ was deposited from InCl_3_ water solution by adding ammonia solution. After washing, the obtained precipitate was dried at 50 °C for 24 h and then annealed at 300 °C for 24 h.

#### 3.1.2. Synthesis of Heterobimetallic Ru(II) Complexes

2-Substituted-1H-imidazo[4,5-f][1,10]phenanthroline compounds **1**, **2**, and the corresponding ruthenium (II) heteroleptic complexes **3** and **4** were prepared as shown in Figure 10. The result of the condensation reaction of the 1,10-phenathroline-5,6-dione with crown-containing aldehyde was the production of ligands **1** and **2** as beige crystalline compounds with good yields (Appendix A) [84]. In the next step, equimolar amounts of ligand **1** or **2** and cis-bis(2,2′-bipyridine)-dichlororuthenium(II) hydrate were kept in ethanol at 80 °C in sealed ampoule under argon for 8 h to prepare the corresponding heteroleptic complexes **3** and **4**. After complete reaction, the crude complexes were purified by column chromatography on aluminum oxide. The novel compounds **2**–**4** were unambiguously characterized by 1H and 13C NMR, ESI MS spectrometry, and elemental analysis (Appendix A). Ligand 1 is described in more detail in [85].

To obtain bimetallic complexes, **3** or **4** were mixed with an equimolar amount of AgClO_4_, Pb(ClO_4_)_2_, or Cu(ClO_4_)_2_ in acetonitrile. The mixtures were then stirred at room temperature for 8 h in darkness to avoid possible photodegradation. Then, bimetallic complexes were precipitated by excess addition of dry diethyl ether. The solid was isolated by filtration and dried under reduced pressure.

For the coordination of the Ag^+^, Pb^2+^, and Cu^2+^ cations in heterobimetallic Ru(II) complexes, fragments of crown ethers were introduced into the ligands: dithia-18-crown-6 for Ag^+^ and Pb^2+^ and azadithia-15-crown-5 for Cu^2+^ (Figure 11a,b) The composition and size of crown ethers were selected for target cations. The aza-containing crown ether with an intermediate Lewis base will firmly bind a copper cation—an intermediate Lewis acid. For silver and lead cations—soft acids, soft Lewis bases are needed. For this, sulfur atoms are well suited. In addition, these cations are larger than Cu^2+^, so 18-membered crown ether was selected. The synthesized heterobimetallic Ru(II) complexes with Ag^+^, Pb^2+^, and Cu^2+^ cations were shortly named as RA, RP, and RC, respectively. The detailed elemental analysis results can be found in the Appendix A.

#### 3.1.3. Synthesis of Hybrid Materials

Hybrids based on nanocrystalline In_2_O_3_ and heterobimetallic Ru(II) complexes were obtained by impregnation. Organic complexes were dissolved in acetonitrile, then 10 µL of the corresponding solution was added dropwise to a weighed portion of the semiconductor oxide, and the suspension was stirred with a glass rod to obtain a uniform distribution of the solution on the surface of the oxide matrix, waiting for the complete evaporation of the solvent each time. The concentration of the solution was chosen so that the content of the Ru in the obtained hybrids was equal to 0.7 mol%.

### 3.2. Materials Characterization

Structural characteristics were investigated by X-ray diffraction in a DRON diffractometer (CuKα radiation, *λ* = 1.5418 Å) and by Raman spectroscopy in an i-Raman Plus spectrometer (BW Tek, Newark, DE, USA). The average size of the crystallites was calculated using Scherrer’s Formula (18):(18)$$d=\frac{k\cdot \lambda }{\beta \cdot \cos\Theta }$$
where *d* is the mean crystallite size (nm), *k* is a dimensionless shape factor and is about 0.9, *λ* = 1.5406 Å is the X-ray wavelength, *β* is the line broadening at the half of maximum intensity, and Θ is the Bragg angle.

The specific surface area was measured by the BET model (Brunauer, Emmett, Teller) in a Chemisorb 2750 instrument (Micromeritics, Norcross, GA, USA). A weighed portion of the powder (~100 mg) was placed in a flowing quartz test tube. The sample was kept in a helium flow (50 mL/min) at a temperature of 200 °C for 1 h to remove adsorbed impurities from the surface. The measurements were carried out in a flow of an N_2_/He mixture (12 mL/min), and the test tube was cooled to liquid nitrogen temperature, recording the absorption of nitrogen from the carrier gas.

The determination of the elemental composition was performed using X-ray fluorescence analysis on an M1 Mistral spectrometer (Bruker). The surface morphology and elemental distribution were characterized using scanning electron microscopy (SEM) and energy-dispersive X-ray spectroscopy (EDS) mapping with a Zeiss NVision 40 (Carl Zeiss, Oberkochen, Germany) microscope. FTIR spectra were recorded on a Perkin Elmer Spectrum One Fourier transform spectrometer in transmission mode in the range of 4000–400 cm^−1^. Samples were mixed with KBr (Aldrich, for spectroscopy) and then pressed into tablets. The absorption spectra of heterobimetallic Ru(II) complexes were recorded on a Cary 300 spectrometer. The diffuse reflectance UV−Vis spectra of the obtained powders were recorded on a Perkin-Elmer Lambda-950 spectrometer in the range of 200–800 nm.

NMR spectra were recorded on INOVA-400 and Bruker Avance 400 spectrometers, (400.13 and 100.13 MHz frequency for 1H and 13C, respectively). Chemical shifts are given with respect to the residual proton signal of the used solvent, i.e., 1.94 ppm for CD_3_CN, 2.49 ppm for DMSO-*d*_6_, 3.31 ppm for methanol-d4, ^13^C: 1.32 ppm for CD_3_CN, 39.52 ppm for DMSO-*d*_6_, 49.00 ppm for methanol-d4). ESI mass spectra (ESI-MS) were acquired on a Finnigan LCQ Advantage tandem dynamic mass spectrometer (USA) equipped with a mass analyzer with an octapole ionic trap, a MS Surveyor pump, a Surveyor autosampler, a Schmidlin-Lab nitrogen generator (Germany), and a system of data collection and processing using the X Calibur program, version 1.3 (Finnigan). The mass spectra were measured in the positive ion mode. Samples in MeCN were injected directly into the source at a flow rate of 50 µL min^−1^ through a Reodyne injector with a loop of 20 µL. The temperature of the transfer capillary was 150 °C, and the electrospray needle was held at a potential of 4.0 kV.

The sensor properties were tested using laboratory-made equipment with a flow chamber at a fixed temperature (T = 25 °C). Electronic mass flow controllers (RRG-12) were used to dilute the test gas and to obtain the desired concentration. The air was purified by a pure air generator (Granat-Engineering Co., Ltd., Moscow, Russia) and was used as background and carrier gas. The concentrations of NO and NO_2_ in gas mixtures were additionally verified with a Teledyne API N500 CAPS NO_X_ Analyzer. Specially designed microhotplates were used for sensor measurements: Al_2_O_3_ substrate covered with Pt electrodes was used for gas sensor measurements. A blue light-emitting diode (*λ*_max_ = 470 nm) with periodic light irradiation was used for photoactivation. The exposure time of the on and off LED was 2 min each.

## 4. Conclusions

Three new heterobimetallic Ru(II) complexes with Ag^+^, Pb^2+^, or Cu^2+^ cations were synthesized and their optical characteristics were studied. The heterobimetallic Ru(II) complexes with Cu(II), Ag(I), or Pb(II) as a second binding cation in the azadithia crown ether moiety were deposited onto the surface of the In_2_O_3_ matrix by impregnation.

It was shown that surface sensitization with heterobimetallic Ru(II) complexes can be used for control of the In_2_O_3_ gas sensor properties. The combination of different properties in the resulting hybrid materials made it possible to eliminate two significant drawbacks of gas sensors—high power consumption and selectivity. Indium oxide, used as a matrix, is an excellent candidate as a sensitive layer for gas sensors because it is chemically stable and has free electrons in the conduction band, surface oxygen vacancies, and active chemisorbed oxygen. The photosensitizer made it possible to reduce the operating temperature of the sensor to room temperature. The second cation plays a critical role in the selectivity toward different gases. Hence, the In_2_O_3_+RC sample had the highest sensor signal toward NH_3_ due to the specific adsorption of NH_3_ molecules on copper cations. Moreover, copper is the stronger Lewis acid compared to silver and lead. This provision can lead to an increase in the sensitivity of copper to ammonia. On the other hand, Cu^2+^ ions are proposed to be active sites for reduction of nitrogen oxides to N_2_. This fact leads to a significant decrease in the sensor signal toward NO_2_ and NO gases.

## Figures and Tables

**Figure 1 molecules-27-05058-f001:**
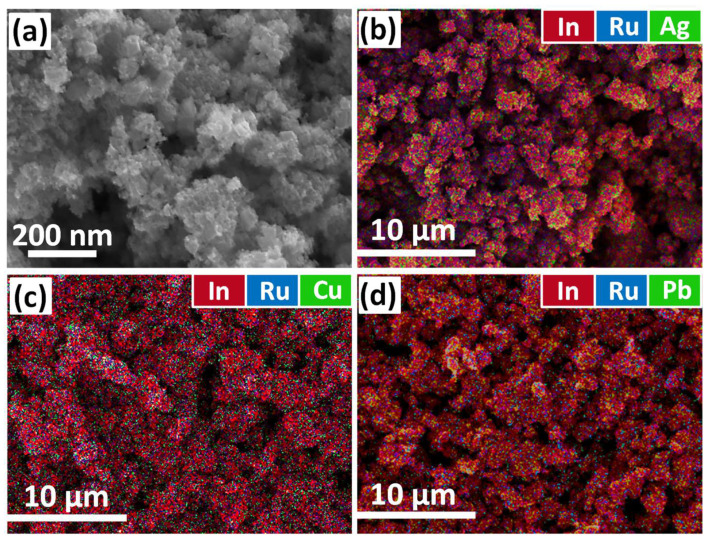
SEM image of the In_2_O_3_ (**a**) and EDS mapping of the various elements (red—In, blue—Ru, green—second binding cation) in hybrid materials (**b**–**d**).

**Figure 2 molecules-27-05058-f002:**
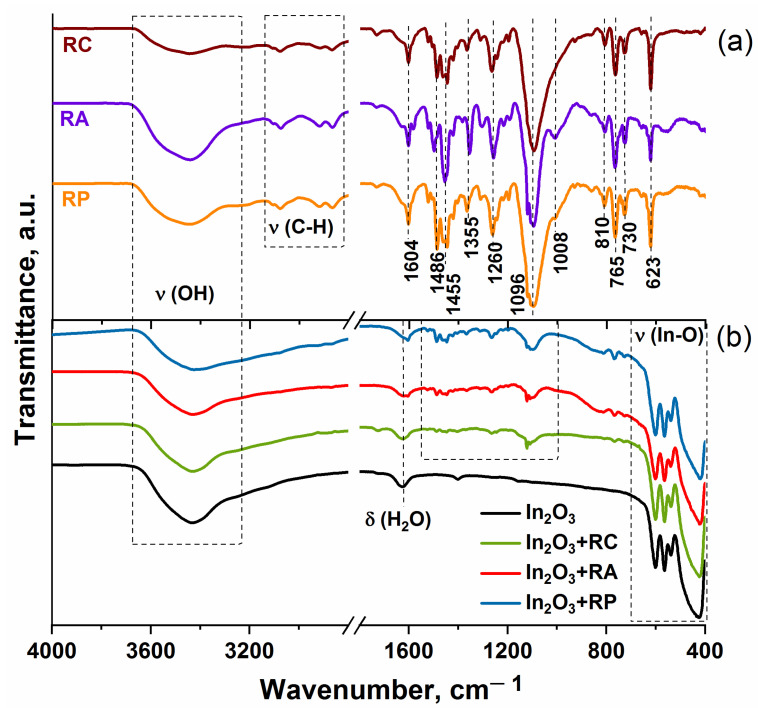
FTIR spectra of heterobimetallic Ru(II) complexes (**a**), pure In_2_O_3_, and hybrid materials (**b**).

**Figure 3 molecules-27-05058-f003:**
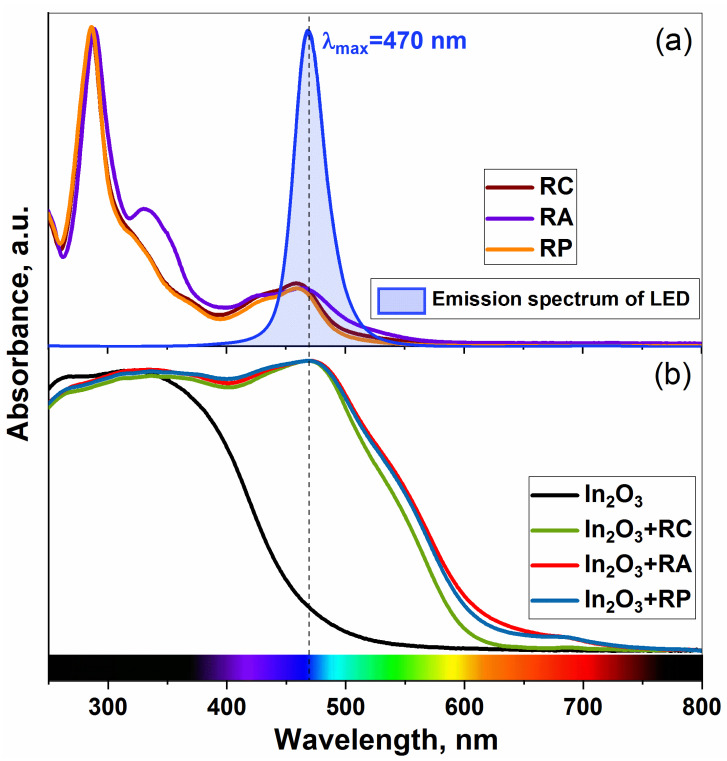
Optical absorption spectra of heterobimetallic Ru(II) complexes (**a**), pure In_2_O_3_, and hybrid materials (**b**). The emission spectrum of the blue LED is also superimposed (**a**).

**Figure 4 molecules-27-05058-f004:**
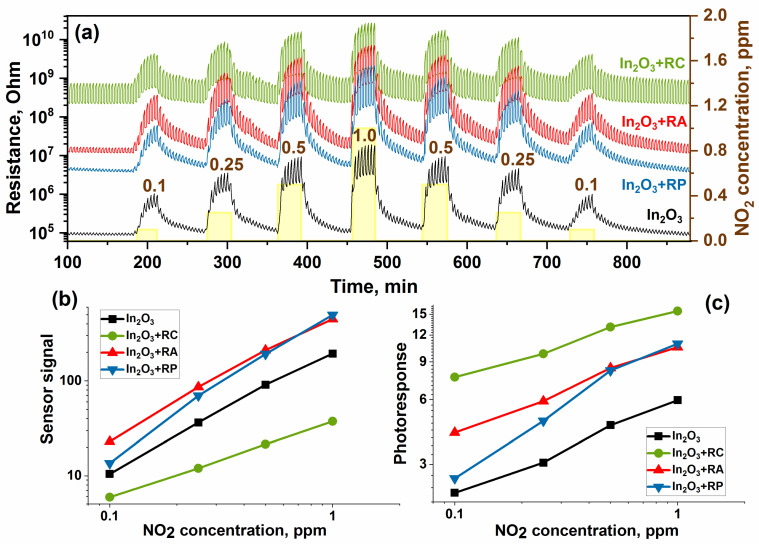
Change in resistance (**a**), correlation of the sensor signal (**b**), and photoresponse (**c**) versus NO_2_ concentration of the samples.

**Figure 5 molecules-27-05058-f005:**
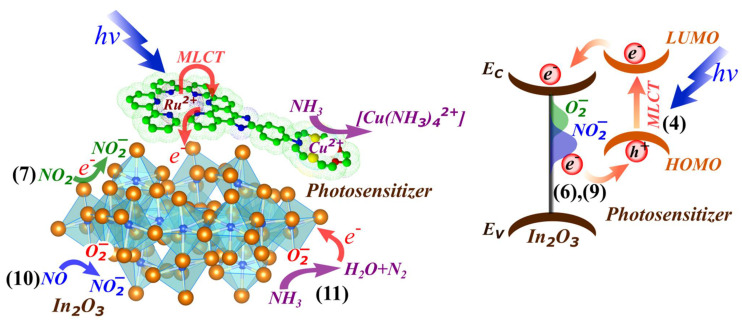
Schematic illustration of the gas sensing mechanism under photoactivation. The numbers in parentheses indicate the ongoing reactions.

**Figure 6 molecules-27-05058-f006:**
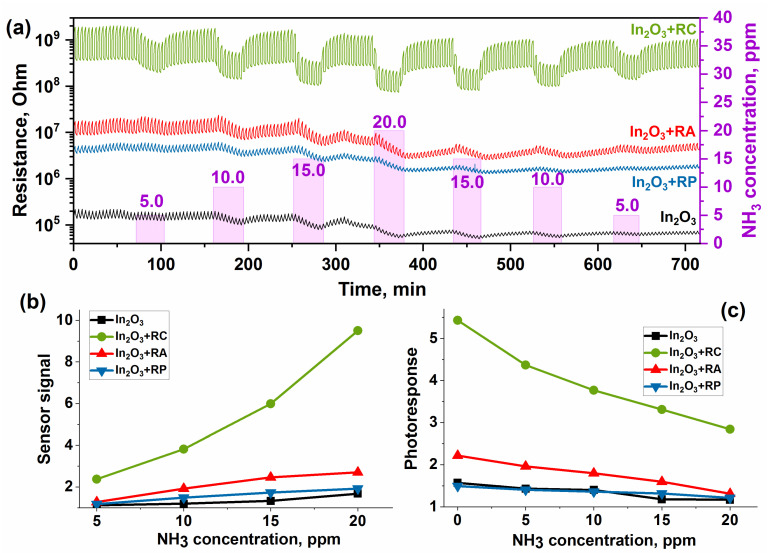
Change in resistance (**a**), correlation of the sensor signal (**b**), and photoresponse (**c**) versus NH_3_ concentration of samples.

**Figure 7 molecules-27-05058-f007:**
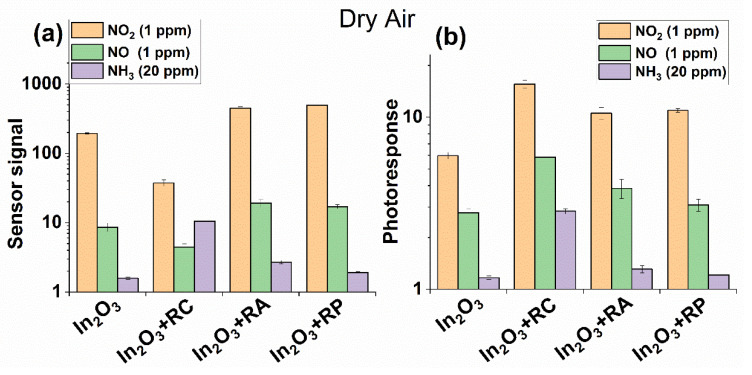
Comparison of the sensor signal (**a**) and photoresponse (**b**) of the samples towards different concentrations of NO_2_, NO, and NH_3_ at room temperature in dry air.

**Figure 8 molecules-27-05058-f008:**
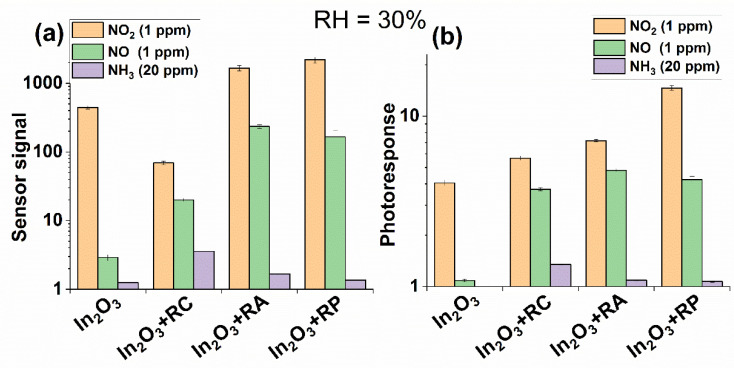
Comparison of the sensor signal (**a**) and photoresponse (**b**) of the samples towards different concentrations of NO_2_, NO, and NH_3_ at room temperature with RH = 30%.

**Figure 9 molecules-27-05058-f009:**
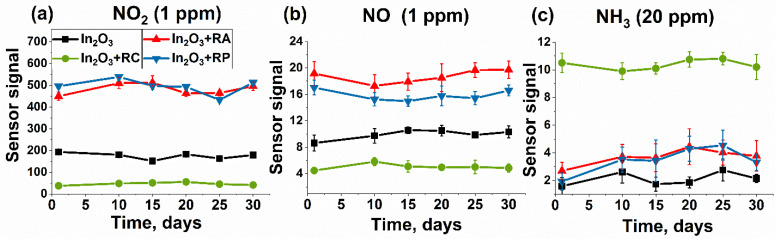
Stability of the sensors for 30 days toward 1 ppm NO_2_ (**a**), 1 ppm NO (**b**), and 20 ppm NH_3_ (**c**) at room temperature under periodic light illumination.

**Figure 10 molecules-27-05058-f010:**
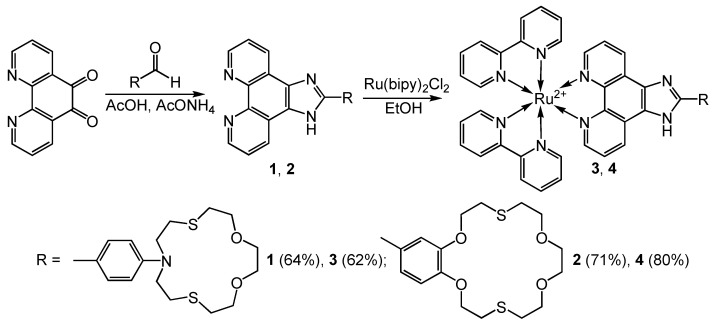
Scheme of the synthesis of ditope ligands.

**Figure 11 molecules-27-05058-f011:**
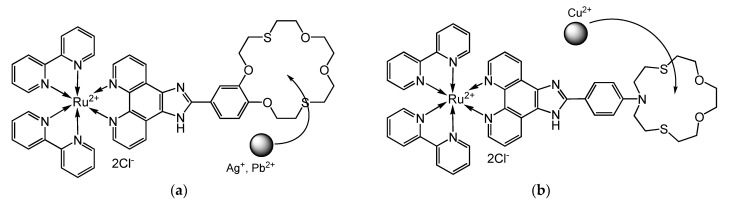
The structures of heterobimetallic Ru(II) complexes with (**a**) Ag^+^, Pb^2+^, and (**b**) Cu^2+^ cations.

**Table 1 molecules-27-05058-t001:** Elemental composition of hybrid materials.

Sample	[Ru]/([Ru] + [In] + [M] *), mol%	[Cu]/([Ru] + [In] + [M] *), mol%	[Ag]/([Ru] + [In] + [M] *), mol%	[Pb]/([Ru] + [In] + [M] *), mol%
In_2_O_3_ + RC	0.68 ± 0.01	0.47 ± 0.01	-	-
In_2_O_3_ + RA	0.71 ± 0.01	-	0.52 ± 0.01	-
In_2_O_3_ + RP	0.77 ± 0.01	-	-	0.57 ± 0.01

Note: [M] ***** = Cu, Ag, and Pb, respectively.

**Table 2 molecules-27-05058-t002:** Assignment of FTIR spectra vibrational modes [34,35,36,37].

Wavenumber, cm^−1^	Vibrational Mode
623, 765	γ (CH), ν (Ru–N)
730	ρ (CH_2_), ν (Ru–N)
810	ν (C–C)
1008	ν (C-O)
1096	ν (C-O), δ (CH) + ν (ring)
1260	ν (C=O) + ν (C=C) + δ (CH)
1355	ν (C=C) + δ (CH)
1455	δ (CH) + ν (C=N)
1486	ν (C–C)
1604	ν (C=N), ν (C–C)
2863, 2917, 3074	ν (C-H)
3450	ν (O-H), ν (N-H)

**Table 3 molecules-27-05058-t003:** Recent works comparing light-activated (with LED) NH_3_, NO, and NO_2_ gas sensor performances at room temperature for different pure and composite materials. Sensor signals were recalculated as:

Material	Wavelength, nm	Incident Irradiance, mW/cm^2^	Sensing Gas	Concentration, ppm	Sensor Signal, %	Refs.
Sulfur-hyperdoped silicon	White LED	0.74	NH_3_	20.0	20,100	[52]
WO_3_ nano-needles	325	0.4	NH_3_	20.0	66.6	[53]
WS_2_ microflakes	365	-	NH_3_	10.0	240	[54]
2DPI/In_2_O_3_ composite	365	-	NH_3_	10.0	950	[55]
CuPc-loaded ZnO nanorods	600–622	0.15	NH_3_	20.0	600	[29]
In_2_O_3_/heterobimetallic complex Ru(II)-Cu(II)	470	8.0	NH_3_	20.0	950	Present study
TiO_x_ nanodots	310	0.3	NO	10.0	31.0	[56]
Au-ZnO nanocomposite	550	-	NO	2.0	194	[57]
In_2_O_3_ nanostructure	365	3.2	NO	2.0	1200	[58]
Nano-porous organic diodes	365	40	NO	1.0	233	[59]
In_2_O_3_/Ru(II) heteroleptic complex	470	8.0	NO	1.0	3000	[22]
In_2_O_3_/heterobimetallic complex Ru(II)-Ag(I)	470	8.0	NO	1.0	1815	Present study
Mesoporous In_2_O_3_	400	-	NO_2_	5.0	900	[60]
In_2_O_3_	385	1.0	NO_2_	8.0	17,900	[61]
WO_3_	590	340	NO_2_	0.16	820	[62]
Au/ZnO	365	1.2	NO_2_	5.0	455	[63]
Al/TiO_2_/Al_2_O_3_/p-Si	254	-	NO_2_	20	11.5	[64]
ZnS-core/ZnO-shell nanowires	254	1.2	NO_2_	1.0	339	[65]
Bi_2_O_3_-core/ZnO-shell nanobelt	254	1.2	NO_2_	1.0	227	[66]
N-719 dye/ZnO hybrid	480	370	NO_2_	1.25	143	[67]
CuO/ZnS nanowire	365	2.2	NO_2_	5.0	955	[68]
CdSe QD@In_2_O_3_	535	20	NO_2_	1.6	10^6^	[69]
CdSe QD@ZnO	535	20	NO_2_	1.6	3000	[69,70]
ZnO/In_2_O_3_ composite	365	25	NO_2_	5	221	[71]
Au/MoS_2_	365	-	NO_2_	2.5	30	[72]
ZnO	455	5	NO_2_	0.025	20	[73]
WS_2_-decorated rGO	430	0.66	NO_2_	1.0	21	[74]
ZnO/(ZnSe(shell) @ CdS(core)) composite	535	20	NO_2_	2.0	6900	[75]
WO_3_	365	8	NO_2_	5.0	11,300	[76]
Perylenediimide-sensitized SnO_2_	400–700	-	NO_2_	0.5	12,900	[77]
Polypeptide-assisted ZnO nanorods	365	-	NO_2_	10.0	400	[78]
Au-ZnO nanorods	495	50	NO_2_	1.0	109	[79]
Ag- ZnO heterostructure	470	75	NO_2_	1.0	150	[80]
MoS_2_/ZnO nanohybrid	365	0.3	NO_2_	0.5	2310	[81]
ZnO/CsPbBr_3_ NCs	470	8.0	NO_2_	2.0	30,000	[82]
In_2_O_3_–ZnO nanotubes	365	1.95	NO_2_	0.5	3170	[83]
In_2_O_3_/Ru(II) heteroleptic complex	470	8.0	NO_2_	1.0	1.75 × 10^5^	[23]
In_2_O_3_/heterobimetallic complex Ru(II)-Pb(II)	470	8.0	NO_2_	1.0	49,500	Present study

## Data Availability

The data presented in this study are available upon request from the corresponding author. The data are not publicly available due to privacy reasons.

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
