# Peer review of "Heterobimetallic Ru(II)/M (M = Ag+, Cu2+, Pb2+) Complexes as Photosensitizers for Room-Temperature Gas Sensing"

_molecules, 2022, doi:10.3390/molecules27165058_

Round 1

Reviewer 1 Report

The design of new sensing nanomaterials based on In2O3 activated by heterobimetallic 2-Substituted-1H-imidazo[4,5-f][1,10]phenanthroline Ru(II) complexes with Ag(I), Pb(II) or Cu(II) is reported in the current article. The obtained materials were carefully analyzed x-ray fluorescent analysis, FTIR spectroscopy and optical absorption spectroscopy. Authors demonstrated the reported materials are sensitive and selective bimodal luminescent and resistive sensors on NO2, NO, and NH3 gases. The sensitivity of the obtained materials is comparable with the light-activated gas sensors reported previously in the literature. Also, authors studied the effect of the second binding cation on gas sensor properties of hybrid In2O3/Ru(II)-M (M= Ag(I), Pb(II) or Cu(II)) heterobimetallic complexes and propose the sensing mechanism and explained the selectivity of different transition metal ion compounds toward different gases. The article is well written, text clear and easy to read, data is clearly presented, and the conclusions are supported by data. This work is interesting for broad range of readers and makes a significant contribution in materials sciences and analytical chemistry. The authors properly revised the manuscript according to the reviewers’ comments, and now, in my opinion, the article can be accepted in present form.

Reviewer 2 Report

The manuscript has been improved

Reviewer 3 Report

This Manuscript is suggested to publish in this journal now.

This manuscript is a resubmission of an earlier submission. The following is a list of the peer review reports and author responses from that submission.

Round 1

Reviewer 1 Report

In this paper, the authors reported the preparation of heterobimetallic Ru(II) complexes with various cations of Ag2+, Pb2+ and Cu2+ for In2O3 based gas sensors. Furthermore, the gas sensing performance of the sensors based on hybrid materials was investigated toward NO2, NO, and NH3 at room temperature under periodic blue LED irradiation. Although, the second binding cation in Ru(II) heterobimetallic complexes plays a critical role in selectivity toward different gases, there are several issues to be addressed in terms of material characterizations and investigations of sensing mechanism before this manuscript can be considered for publication (Major revision).

 1. In the introduction, there is no direct relationship between COVID-19 and gas sensing. Related descriptions should be added.

2. How can the authors obtain the average grain size of In2O3 matrix is 13-14 nm? Please give the specific calculation formula.

3. Please add the SEM images of obtained materials.

4. Due to the specific surface area of gas sensing materials influencing the gas sensing properties, it is necessary to add the specific surface areas of different hybrid materials (In2O3+RC, In2O3+RA, and In2O3+PR). And please give the detailed results of the BET measurements.

5. Generally, the EDS mapping is more suitable for the analysis of the elemental composition of hybrid materials. I suggest the author to add this test.

6. Why did the authors choose 0.7 mol% as the content of the Ru in hybrids? Please explain the reason.

7. Some recent literatures on the explanation of the gas sensing mechanism recommend to review in the current manuscript, such as The Chemical Record, 2021, 21, 1-2; Sensors & Actuators: B. Chemical, 2022, 354, 131206, etc.

8. From Fig. 3 and 4, the second cation in Ru(II) heterobimetallic complexes can influence on the response signal of gas sensors toward different gases. Please explain the gas sensing mechanism based on different second cation (Ag2+, Pb2+ and Cu2+).

Reviewer 2 Report

The design of new nanomaterials based on In2O3 activated by heterobimetallic 2-Substituted-1H-imidazo[4,5-f][1,10]phenanthroline Ru(II) complexes with Ag(I), Pb(II) or Cu(II) is reported in the current article.  The obtained materials were carefully analyzed x-ray fluorescent analysis, FTIR spectroscopy and optical absorption spectroscopy. Authors demonstrated the reported materials are sensitive and selective luminescent and resistive sensors on NO2, NO, and NH3 gases. The article is well written, data is clearly presented, and the conclusions are supported by data. This work is interesting for broad range of readers and makes a significant contribution in materials sciences and analytical chemistry. In my opinion, the article can be accepted in present form.

The design of new sensing nanomaterials based on In2O3 activated by heterobimetallic 2-Substituted-1H-imidazo[4,5-f][1,10]phenanthroline Ru(II) complexes with Ag(I), Pb(II) or Cu(II) is reported in the current article. The obtained materials were carefully analyzed x-ray fluorescent analysis, FTIR spectroscopy and optical absorption spectroscopy. Authors demonstrated the reported materials are sensitive and selective bimodal luminescent and resistive sensors on NO2, NO, and NH3 gases. Also, authors studied the effect of of the second binding cation on gas sensor properties of hybrid In2O3/Ru(II)-M (M= Ag(I), Pb(II) or Cu(II)) heterobimetallic complexes and propose the sensing mechanism and explained the selectivity of different transition metal ion compounds toward different gases. The article is well written, text clear and easy to read, data is clearly presented, and the conclusions are supported by data. This work is interesting for broad range of readers and makes a significant contribution in materials sciences and analytical chemistry. In my opinion, the article can be accepted in present form.

Reviewer 3 Report

The authors report on the influence of the second binding cation on light-activated gas sensor properties of the hybrids based on nanocrystalline In2O3 and Ru(II) heterobimetallic complexes. Many hybrid based In2O3 sensor for ammonia detection at room temperature with very good sensing properties  have been already reported (10.1016/j.snb.2022.131918;   10.1016/j.ceramint.2021.11.209)  as well as for NO2 detection at room temperature (10.1016/j.snb.2015.04.119). In this paper,the sensing properties are not well investigated. sensor response and recovery times should be determined,the long term stability should be investigated as well as the effect of humidity which is a very important factor when the sensor operates at room temperature. The results should be clearly compared to other results reported in the literature on ammonia, NO2 and NO detection in order to show the superiority of this sensor. The sensing mechanism should be illustrated with a diagram. The manuscript contains many spelling mistake (line 247, line 238 (ref 22 should be replaced by the name of the main author before adding the reference and this can be seen in other part in the manuscript)).